# Design of a Small Unmanned Aircraft System for Bridge Inspections

**DOI:** 10.3390/s20185358

**Published:** 2020-09-18

**Authors:** Travis Whitley, Andy Tomiczek, Chad Tripp, Andrew Ortega, Matlock Mennu, Jennifer Bridge, Peter Ifju

**Affiliations:** 1Department of Mechanical and Aerospace Engineering, University of Florida, Gainesville, FL 32611, USA; chadtripp@ufl.edu (C.T.); mmennu@ufl.edu (M.M.); ifju@ufl.edu (P.I.); 2Department of Civil and Coastal Engineering, University of Florida Gainesville, FL 32611, USA; andytomi@ufl.edu (A.T.); jennifer.bridge@essie.ufl.edu (J.B.); 3Department of Computer Science, University of Florida Gainesville, FL 32611, USA; andrew.ortega@ufl.edu

**Keywords:** optical flow, bridge inspection, GPS-denied navigation, computer vision, electrical optical sensor, laser rangefinder, obstacle avoidance

## Abstract

Bridge inspections are an important procedure for maintaining the infrastructure vital to our economy and well-being. The current methodology of utilizing specialized equipment such as snooper trucks and scaffolding to support manned-inspections poses a significant financial cost, disrupts traffic, and is dangerous to the inspectors and public. The advent of unmanned aerial systems (UAS), more commonly called drones, presents a practical solution that promises reduced cost, enhanced safety, and is significantly less intrusive than previous methodologies. Current limitations in the implementation of UAS include the reliance on a skilled operator and/or the requirement for a UAS to operate in a cluttered, GPS-denied environment. A solution to these challenges is presented in this paper by utilizing commercial off-the-shelf (COTS) hardware including laser rangefinders, optical flow sensors, and live video telemetry. Included in the system is the obstacle avoidance equipped drone and a ground station intended to be manned by a pilot and bridge inspector. The proposed custom-fabricated UAS was implemented during eight inspections of Florida Department of Transportation (FDOT) bridges. The UAS was able to navigate under GPS-denied and obstacle-laden bridge decks with position-hold performance comparable to, if not better than, a COTS unit in an unobstructed environment. The position hold capability maintained an altitude of ±12.8 cm with a horizontal hold of ±435 cm. Details of the hardware, algorithm development, and suggestions for future research are discussed in this paper.

## 1. Introduction

The goal of this paper is to document the development of a proposed unmanned aerial system (UAS) that can be used to survey the underside of bridge decks. Current bridge inspection methodologies involve using expensive equipment, including snooper trucks, that can be hazardous to operate and disruptive to the surrounding traffic [1,2]. A promising solution to this is the use of UAS which present a viable low cost alternatives that do not pose a direct risk to the operators and the public, all without hindering the flow of traffic [3,4]. In order to be a viable alternative, there are many problems that UAS have to overcome when it comes to under bridge inspections such as unreliable GPS access and operating in a cluttered environment [5,6,7]. One attempt to overcome this problem is to fly the vehicle to the side of the bridge and capture images from an oblique angle [5,6] at the cost of picture quality while other implementations involves relying on pilot skill to navigate COTS systems around these obstacles [4,8,9]. Based off the current findings, there is a need for a semi-autonomous solution which can allow bridge inspectors themselves to operate the aerial vehicle, thus minimizing the cost of hiring a skilled worker such as a drone pilot. Pilot sensory overload is common when operating in such environments and additional stabilization/obstacle avoidance technologies allow for fewer collisions and reduced operator workload while also allowing for more efficient coverage of the structure.

In previous work, computer vision—specifically visual odometry—has been proposed as a candidate for a viable option for GPS-denied navigation [10,11,12]. This application of computer vision for navigation under bridges has yet to be widely adopted. While simultaneous localization and mapping (SLAM) and similar feature-matching algorithms can provide reliable position information to navigate by [1,13], they require significant computational resources to produce a real time solution. These requirements dictate additional equipment, cost, and weight for the aircraft. Other methodologies have employed triangulation and stereovision techniques, but also have similar drawbacks of added equipment and complexity [14,15].

Optical flow, specifically the Kanade-Lucas-Tomasi (KLT) method [16], is a robust solution for navigation with low computational cost [17,18]. In addition, open-source hardware which integrates all the system requirements (such as the PX4FLOW (ETH Zurich, Switzerland) sensor utilized in this project) is already commercially available [19].

In this paper, the authors will elaborate on the hardware solution utilized in our proposed system, which includes a sensor enhanced drone and ground station. The control algorithm will be addressed, along with the assessment of performance. To evaluate the performance of the proposed system, Digital Image Correlation (DIC) was employed. DIC is a non-contact measurement technique that was developed at the University of South Carolina [20,21,22]. This methodology uses image mapping techniques to determine changes between successive images and translates them into displacements and strains at high accuracy. DIC requires a speckled pattern with high contrast to correlate subsets of the image.

During the research program to develop the system presented in this paper, numerous iterations of hardware and algorithms were built, calibrated and tested in a real-world environment. Some concepts that seemed to perform in the lab setting were not robust enough under bridge decks where wind, poor lighting, absence of GPS signal, and significant clutter/complex structural features prevail. For brevity, those iterations will not be discussed in detail. Additionally, the system was developed with Federal Aviation Administration (FAA) compliance at the time of the project. FAA required a three-person flight crew during the project development and evaluation phase.

## 2. Hardware Solution

### 2.1. Airframe

Two overriding requirements of the proposed system: (1) the ability to stabilize the drone in a GPS-denied environment, and (2) a camera positioned above the drone to view the underside of the bridge deck, drove the overall design. Additionally, pilot and inspector sensory overload was a primary concern during system development. A typical starting point for many UAS projects is to start with a COTS airframe and control system. Since most systems rely on GPS for position hold and relatively low-resolution video cameras slung under the drone, the authors decided a custom hardware solution would better meet the application design requirements. The hardware configuration can be seen in Figure 1. The airframe that was chosen was a quadrotor design which allowed the pilot to hover in place and navigate around obstacles such as concrete piers and girders. The airframe utilized carbon fiber construction to minimize vehicle weight to maximize flight time and agility. Lithium polymer battery chemistry was selected as it provides the highest power/energy density for the aircraft power requirement. For visual inspection, a 24 megapixel A6000 digital SLR camera (Sony, Minato City, Tokyo, Japan) was used in conjunction with a variable zoom lens, wireless video link, and remote controls for image capture and video recording. Numerous other cameras were evaluated in the field, but the combination of on-the-fly changes from high resolution video to 24 megapixel stills with optical zoom ranges that matched the under-bridge setting, and low light/automated light level adjustments led to the decision to incorporate the Sony A6000. A variable zoom lens was utilized to allow the operator to zoom in on defects without needing to fly closer, as suggested in previous studies [5,6]. The camera system was then mounted to a gimbal system utilizing stabilized servo control, which was manipulated wirelessly over a 2.4 GHz RC link. The camera was mounted above the aircraft as previous studies determined this is the optimal placement for under bridge deck inspections [5,6,7]. The A6000 was meant to be controlled by an FDOT bridge inspector, while the pilot utilized a first-person view (FPV) video with one axis of gimbal rotation (front to back) for controlling the drone. More than one axis of rotation of the FPV camera resulted in pilot disorientation. The FPV camera was also selected to help the pilot navigate around horizontal obstacles [23,24]. To round out the sensor/control sweet, an optical flow sensor, laser range finder, PixHawk autopilot (3D Robotics, Berkeley, CA, USA), GPS and video transceivers were incorporated, as can be seen in Figure 1. Foam was added on the carbon fiber landing gear in case of an unintended water landing since most of the bridges that were inspected spanned bodies of water.

### 2.2. Ground Station

The proposed ground station, shown in Figure 2, was comprised of a tripod system containing two displays and a laptop for control. The two displays were for the inspector and the pilot, each with corresponding controllers. The inspector’s display shows a live feed of the inspection camera with gimbal, zoom, video/still controls, while the pilot’s display shows the FPV camera fixed to the airframe with pitch stabilization. The inspector’s controller can change the yaw of the inspection camera relative to the airframe for a complete 360 degree range while also controlling the stabilized pitch of the camera allowing for 135 degrees of range. The controller also allows for manipulation of the optical zoom of the inspection camera in addition to camera functions including video recording and high-resolution image capture. The pilot’s controller allows for positioning the aircraft in addition to changing the stabilized pitch angle of the FPV camera which broadcasts to his display in real time. There are also switches for the pilot to toggle flight mode changes of the aircraft itself. The ground station computer consists of a laptop connected to a 900 MHz wireless modem which displays telemetry from the aircraft’s autopilot. This information is used to monitor the battery life of the aircraft and other critical flight information. Additional items of the ground station included the telemetry units for the cameras, a deep-cycle lead-acid battery, voltage inverter, and sun shields for the monitors. While this setup was sufficient for the scope of this research, a more compact form factor would be required for a commercial offering.

## 3. Algorithm Development

### 3.1. GPS Denied Navigation

To determine the global positioning of the aircraft from rangefinder data and optical flow measurements, kinematic equations need to be derived to convert these quantities to velocities in a global reference frame. These global velocities can then be integrated into a relative position vector as we will be utilizing a position based visual servo control algorithm [10]. For the derivation, the global reference frame will be defined as {*Nx, Ny, Nz*}, the body fixed coordinate system of the camera and the airframe is defined as {*Cx, Cy, Cz*}. The location of the origin of the global reference frame is defined as O, the location of the focal point of the optical flow camera is defined as F, and the intersection of the optical axis of the optical flow camera with the overhead scene to be tracked is defined as P, as shown in Figure 3. 

The relationship between the three points are defined in Equation (1) where we desire to formulate the location of point f in the global horizontal plane defined by *Nx* × *Ny*:(1)$$r_{p/o}=r_{f/o}+r_{p/f}$$

$r_{p/f}$ can be determined using the standoff distance calculated from a combination of the rangefinder and the aircraft INS while the velocity of p with respect to O can be related to the optical flow measurements. Solving for $r_{f/o}$ we obtain Equation (2). We do this since the vector $r_{f/o}$ represents the location of the aircraft in the global reference frame:(2)$$r_{f/o}=r_{p/o}-r_{p/f}$$

By taking the time derivative of Equation (2) and applying the transport theorem [25] we can obtain Equation (3) which relates the velocity of the feature points and gyro measurement ωCN, to the global velocity. By getting the equation into the velocity domain, we are able to utilize velocity estimates from the optical flow sensor to describe the velocity of the aircraft in the global frame which can then be integrated for position:(3)vf/oN=vp/oN−(vp/fC+ωCN×rp/f)

Since feature points are assumed to be stationary, we can assume they have a zero velocity with respect to the origin of the global reference frame which is represented as $v_{p/o}^{N}$ in Equation (3). In addition, since we are calculating an average velocity of all the feature points within the field of view of the camera, we can assume the average velocity is being measured at where the camera’s optical axis intersects the ceiling. With these assumptions, we can simplify Equation (3) into Equation (4):(4)vp′/pN=−vp/fC−ωCN×rp/f

The feature point velocity is computed in the pixel space of the camera. To relate these velocities to the global reference frame, a pinhole model of the camera is used as seen in Figure 4. Using this model, we can relate the position of a pixel point to the physical world using Equation (5):(5)$$P_{i}\left(\frac{n}{f}\right)=P_{f}$$

Here, $P_{f}$ represents the coordinates of the feature point in the camera reference frame, $P_{i}$ is the location of the corresponding feature point in the image reference frame, n is the normal distance from the optical axis to the feature point, and f is the focal length. We can take the time derivative of Equation (5) to give us the velocity of the feature point in the camera frame. Once the derivative is taken, the equation can further be simplified since the focal length does not change with time (only valid for fixed focal length cameras) and the velocities of the feature points are averaged at the center of the image giving us Equation (6):(6)$$v_{i}\left(\frac{n}{f}\right)=v_{f}=-v_{p/f}^{C}$$

As stated before, the expression ωCN is determined by sensor data obtained directly from a gyro in the form of Equation (7):(7)ωCN=−gxCx−gyCy−gzCz

Substituting Equations (7) and (6) into Equation (4), we can now calculate the global velocity in terms of the average pixel velocities and information from the gyros as shown in Equation (8). This equation can then be integrated with time to determine position:(8)$$v_{f/o}^{N}=\left(\frac{v_{iy}}{f}-g_{y}\right)nC_{x}+\left(\frac{v_{ix}}{f}-g_{x}\right)nC_{y}$$

### 3.2. Vertical Obstacle Avoidance

To maintain a safe distance below the bridge deck, the aircraft utilizes an upward facing laser rangefinder. The rangefinder measurements are first run through a fixed gain Kalman filter before being passed to the aircraft’s obstacle avoidance algorithm. The form of the Kalman filter is shown in Equations (9) and (10):(9)x(n+1)=Ax(n)+Bu(n)+Gw(n)
(10)y(n)=Cx(n)+Du(n)+Hw(n)+v(n)

For this formulation, we assume the standoff distance can be obtained by integrating the climb rate, thus why the state transition matrix A is of the form as shown in Equation (11):(11)$$A=\left[\begin{array}{cc} 1 & dt\\ 0 & 1 \end{array}\right]$$

Since the inputs into the system will not be monitored, the B and D matrices will be zero. We will also assume the process and measurement noise, represented as H and G respectively, effect the measurements independently, thus they will be represented as identity matrices. The output matrix, represented as **C**, will be an identity matrix since the sensors are directly measuring the state of the system. The state vector, x, of the Kalman Filter is the standoff distance and is corresponding velocity, while the output vector, y, describes the standoff distance from the rangefinder itself, s, and the reported climb rate of the air vehicle, *C_R_*_,_ as shown in Equation (12):(12)$$y=\left[\begin{array}{c} y_{1}\\ y_{2} \end{array}\right]=\left[\begin{array}{c} s\\ -C_{R} \end{array}\right]$$

The fixed gain Kalman Filter is used since we assume the system noise will not change with time and to reduce the computational load on the autopilot hardware. This filter allows for the rangefinder distance measurements to be fused with the aircraft’s existing vertical velocity estimates. These vertical estimates are determined from the autopilot’s existing Extended Kalman Filter which fuses the barometer height estimates with a six DOF IMU and three axis magnetometer. The noise covariances are determined from a combination of static test data in addition to adjusting the noise covariances in the field until the desired performance is obtained. Once the system is implemented and tuned correctly, the vertical offset readings (estimated in Equation (9)) are then passed to both the optical flow algorithm and the overhead obstacle avoidance algorithm.

The goal of the overhead obstacle avoidance algorithm is to maintain a prescribed offset from the bottom of the bridge deck, while still allowing the pilot to change altitudes manually. These goals have produced the following flight algorithm as illustrated in Figure 5.

This algorithm breaks the overhead distance into three categories based on a minimum distance and a slowdown distance. When the aircraft standoff distance is within the minimum standoff distance, the aircraft commands a constant descent rate and the pilot is limited such that they can only command a higher descent rate. When the aircraft is above the slow down distance, the pilot has full range of desired climb rates. When the aircraft is between the slow down distance and the minimum standoff distance, the maximum climb rate is linearly reduced based on offset distance until the aircraft reaches the minimum standoff distance, in which case only descent rates can be commanded. The reason for this third category is to minimize overshoot into the underside of the bridge deck. The benefit of this algorithm is the pilot can track the bottom of the bridge deck by always commanding an upward climb rate, but still have normal control when the aircraft is a safe distance from the bridge deck.

### 3.3. Hybrid Altitude Control

While typical autopilot systems for multi-rotors primarily rely on barometric pressure for altitude control, this methodology suffers from barometric drift, defined as the drift in altitude readings coming from uncompensated changes in temperature and Venturi effects on the aircraft’s fuselage. An alternative to this is to use an attitude compensated rangefinder to determine altitude. While this alternative does not suffer from barometric drift, the device can only show an altitude relative to the surface the rangefinder is facing. In the case of bridge inspection, this would cause discontinuous jumps in altitude when traversing over concrete girders. To overcome this, a novel approach of implementing a state machine is used. The algorithm takes the discrete velocities of the aircraft’s barometrically derived altitude (C_R_ in Equation (12)) and compares it to the velocity of the rangefinder altitude derived from the Fixed Gain Kalman Filter described in Equation (9) to identify and eliminate these discontinuous jumps. When the difference between these velocities is above a specified threshold, the system will revert back to utilizing barometric-pressure based altitudes, as illustrated in Figure 6 by simply integrating the vertical velocity of the aircraft obtained from the EKF fused barometer and accelerometer readings, (represented as C_R_). If instead these velocities are within the specified threshold, the Kalman filtered rangefinder velocity is simply integrated and added to the previous altitude. While within this regime, barometer drift is non-existent as the sensor driving the altitude solution is solely the rangefinder itself. In the even the rangefinder returns an out of range reading, the system will revert back to the barometric derived altitude as before since the relative distance between the aircraft and bridge deck is unknown.

## 4. Materials and Methods

### 4.1. Bridge Inspection Test Site

A local, pre-stressed concrete bridge, shown in Figure 7, located in Alachua (FL, USA) was selected as the location to benchmark the UAS for conducting bridge inspections. The bridge was used for real-world testing the system, including the GPS denied stabilization algorithm, the imaging system with pilot and inspector roles, and the flight performance characteristics such as flight duration.

This multi-span bridge was selected due to its close proximity to the University of Florida (UF) campus (less than 30 min drive), the variety of bridge deck types, lighting conditions comparable to those common in the state of Florida, and its relative solitude (the two parallel bridges have a total of ten spans; two over Rachael Blvd. which has a low amount of traffic, four over railroad tracks and four without any roads or tracks underneath). Over a period of a year the team spent over 100 h testing various evolutionary versions of prototype hardware and software. Many algorithm decisions in terms of vehicle control and stabilization, inspector and pilot choreography, camera settings, and PID/Kalman filter tuning settings were made during these testing sessions. Some of the types of activities and outcomes are reviewed in this section.

The first flights with a 3-axis brushless gimbal system demonstrated the system could provide excellent video stabilization in addition to intuitive user control, the weight of the configuration proved to limit the flight endurance of the UAS. Because of this an optimal gimbal configuration was sought to optimize weight while providing sufficient image stability for identifying defects on bridges. The first tradeoff was to switch from a brushless gimbal motor setup to a servo motor system. While the brushless setup provided better image quality, the servo motor setup provided enough stabilization to identify defects while being lighter. Reducing the degrees of freedom of the gimbal system was also investigated as this would reduce the weight of additional motors and hardware. Originally, a roll-pitch configuration was proposed but the controls for the camera were not as intuitive for the operator in addition to dictating a reliance on communication with the pilot to help orient the vehicle. This in turn would add to the cognitive load of the pilot, which was not desired. Instead, a yaw-pitch configuration was settled on as the optimum configuration. The control of the system was straight forward and allowed for control independent from the pilot’s inputs. Another configuration was conducted which looked at the feasibility of running a fixed focal length GoPro camera with a small focal length but while this saved significant weight, the resolution was not enough to identify defects under the bridge.

Overhead obstacle avoidance development was also evaluated at the test site. The first iterations of the algorithm consisted of limiting climbs if an obstacle is within a prescribed distance of the laser rangefinder. It was determined that this implementation was not sufficient as the operator still had to manually descend the UAS to a safe distance below the bridge deck. Variances in commanded altitude, as a result of utilizing a barometric pressure sensor, would sometimes cause the aircraft to overshoot and contact the bottom of the bridge deck. As a result, a provision was added to the algorithm to descend the UAS at a certain rate when an obstacle breaches the standoff distance. Readjusting the target altitude directly based off the difference between the standoff distance of the aircraft was not utilized as the discrete jumps in laser rangefinder readings caused by traversing over the concrete girders would cause the aircraft to constantly shift target altitudes. The stepping caused undesired performance, thus a continuous descent was chosen. To prevent the overshoot issues, it was determined that the climb rate should be linearly reduced based off standoff distance to keep the aircraft from overshooting into the bridge deck and allow the operator to precisely maneuver to a prescribed standoff distance. Another attempt to reduce the stepping was to widen the beam of the laser rangefinder, such that the average depth of a larger area was more continuous in nature. A compromise could not be reached for the particular sensor setup for widening the laser beam as this would result in too dramatic of a reduction in range. Ultrasonic sensors were also investigated, but sensor reliability was determined to be an issue.

The final optimizations from the testing was to the bridge inspection process itself. It was eventually found that the best procedure was to fly to the middle of the bridge deck and ascend to within range of the optical flow sensor. Once in range of the optical flow sensor, the UAS was oriented to the outside girder and begin to fly up and down each girder. During each transition, the aircraft would first take a picture of the pier and endcap before jumping to the next girder and then rotate to look down it. A picture would then be taken before forward flight was resumed. This would continue until the whole bridge deck was traversed. It also provided the ability to test out the generation of a 3D point cloud reconstruction of the bridge using images captured from the inspection camera.

### 4.2. Experiments Conducted to Validate the System

Three tests were performed to validate the bridge inspection UAS’s ability to hold position. The first test was conducted outdoors, in an open area (not under a bridge), to validate the GPS and pressure-based altimeter position holding ability with comparison to a COTS drone. The drone that was chosen for comparison was the S1000+ (DJI, Nanshan, Shenzhen, China) heavy lift vehicle. This vehicle was chosen due to its similar size, ubiquitous nature, and availability in the UF UAS Lab. A second test was performed under a bridge deck, described in the previous section, with the optical flow and range finder algorithm for position hold with no reliance on GPS. A third test was conducted to understand how accurately the bridge inspection drone performed while traversing under one of the bridge girders.

To compare the performance between the optical flow-based navigation system and a COTS GPS navigation system, DIC with stereo cameras was used to measure the 3D motion of the vehicles as a comparison metric. The authors chose this methodology based on the high displacement resolution of the method, the ability to operate under outdoor conditions, and the availability of the system in the UF UAS Lab. To utilize this reference system, a speckled carbon fiber honeycomb sandwich plate measuring 17” by 17” was affixed to the bottom of the aircraft. The average speckle size used in the pattern corresponded to approximately five pixels for the cameras/lenses based on the average distance between the drone and cameras. Two stereo cameras were affixed on each end of a 2 m rigid steel beam. The zoom, aperture, and focus were set to optimize the system for the widest field of view with the aircraft in focus at a 5 m altitude. The stereo cameras were then calibrated using Correlated Solutions Inc., VIC-Snap software.

To document the measurement resolution of the DIC setup, the carbon fiber plate with the speckle pattern was mounted to a vice instrumented with a micrometer translation stage to measure the actual displacement. The carbon fiber plate was then positioned in front of the DIC cameras at a distance of 5 m, which was the typical altitude of the aircraft relative to the cameras during the outdoor trials. Eleven measurements were obtained from the DIC setup of 11 known displacements, as measured from the micrometers, each of 0.10” displacements from each other. These displacements were recorded ten times each and then fitted with linear regression. The data points can be seen in Figure 8.

The displacement measurements were made in the out-of-plane direction (Z direction corresponding to altitude) and the in-plane direction (X-Y direction). It was found that the standard error for the DIC measurements versus out-of-plane for in-plane and out-of-plane measurements came out to be 0.02059 mm and 0.06796 mm respectively. Due to the standard error being so small, the resolution for all the plots obtained from these experiments can conservatively be estimated to be smaller than the line thickness used. The high measurement resolution compared to typical fiducial based stereo measurements is due to the large area of the speckled target plate, and the averaging algorithm used in the DIC software.

For first series of outdoor tests, both the UF and the DJI drones were commanded a stationary position above the center of the stereo pair cameras without pilot intervention. Figure 9 shows photographs of the DJI drone on the ground and in the air above the stereo camera pair, the computer screen of the DIC image capture system, and the UF drone above the camera pair. The autopilots used have built-in logging that were used to compare the aircraft’s estimated and actual positions. The GPS aircraft were flown in an open field, free of obstructions, on a relatively windless day. In the second test, shown in Figure 10, the GPS-denied aircraft flew underneath a bridge between two girders where the ceiling was relatively flat with concrete texture. To synchronize the data logs from the autopilot to the DIC data the aircraft was commanded a 90 degree yaw deflection in both directions as a trigger. The camera capture rate for DIC was set to 10 Hz, the maximum rate available for the cameras. The third test was performed with the optical flow system, which tested the obstacle avoidance of the aircraft. Figure 11 illustrates that in this test, the aircraft was flown back and forth under a concrete girder below a bridge where the autopilot would correct the altitude to navigate around the obstacle.

## 5. Results

The first run of the GPS trial was with the S1000+ as shown in Figure 12. This system was used to demonstrate the position hold performance of a typical COTS system. The aircraft was flown above the DIC setup at an altitude of 5 m and was commanded to hover in place. The aircraft was able to hold an in-plane position of ±720 cm while maintaining an altitude within ±198.9 cm. The estimated position from GPS and the measured position from the DIC system are plotted for displacements in the x and y directions. The initial position when the pilot released control is the origin of the plots. Figure 13A shows three separate 2 min runs, from the point of view of an observer on the ground looking up at the vehicle. Figure 13B shows the altitude hold. Additionally, the estimated and measured velocity of the aircraft were plotted and shown in Figure 14.

The quadcopter used for bridge inspections was flown utilizing a GPS navigation system and barometric pressure for altitude, basically in the same mode as the DJI platform. The aircraft was flown at the same altitude (5 m) and was commanded a position hold over the DIC setup. The aircraft was able to maintain ±487 cm of lateral position hold and maintain altitude within ±96.4 cm as depicted in Figure 15 and Figure 16.

The second series of tests, where the bridge inspection UAS was flown without reliance on GPS, under a flat portion of the bridge deck, was conducted to compare the holding position capabilities as compared to that in the open with GPS reliance. As seen in Figure 17, the position hold performance was comparable to both the GPS COTS system and to the aircraft using GPS navigation in an unobstructed environment. This shows that the system can be used for GPS denied navigation with a similar level of performance of other COTS systems. The following runs produced an average position error of ±507 cm and maintain an altitude within 30.4 cm. Additionally, as seen in Figure 18, there was a similar relationship between estimated and measured velocities as with the DJI platform.

In the final flight tests, to analyze the overhead obstacle avoidance, the aircraft was flown back and forth under a girder, as illustrated schematically in Figure 11. As shown, the aircraft begins descending once under the girder. Once the aircraft has cleared the girder, it begins to ascend to the standoff distance. Figure 19 plots the altitude over a 60 s interval, displaying the standoff distance measured by the laser rangefinder, the estimated altitude from the extended Kalman filter and the measured altitude using DIC. During the 60 s trial, the vehicle was flown back and forth under the girder 3–1/2 times. This demonstrates how the aircraft only begins descending once the rangefinder reports readings under 2 m. In addition, one can observe the two climb rate responses where the aircraft descended at the maximum descent rate and then climbed at a reduced rate. Once the obstacle was cleared, the vehicle began to ascend at the maximum climb rate until it is within the slow down standoff distance. At this point, the aircraft ascends at a slower rate until it reaches the standoff distance.

## 6. Bridge Inspection Summary

To test the capabilities and provide real world feedback, the UF aircraft was used to inspect eight different bridges [3] as shown in Figure 20. At each bridge inspection, at least one certified bridge inspector (CBI) was present to help guide and critique the system. The bridges were selected to encompass a broad variety of construction types and environmental locations. The summary of the bridge inspections and improvements are detailed in Table 1.

After the eight inspections, it was found that the vehicle was able to inspect all but one bridge utilizing optical flow hold, but the bridge construction did play a role in the position hold performance of the system. For example, the system held a tighter position when flying under pre-stressed concrete bridges than steel bridges. This performance variation can be linked to two possibilities. The first possibility is the steel girders could be affecting the magnetometer readings. The magnetometer is needed for determining an accurate heading of the aircraft, which is used to determine position from the optical flow sensor which can explain why a bad heading can cause a poor position estimate. Alternative heading estimation utilizing the optical flow readings themselves has been proposed to overcome this problem [26,27]. The pre-stressed concrete girders also had a different texture than the painted steel girders, that could have provided more feature point for the algorithm to track, thus bolstering performance.

The main deficiency of the system was determined to be performance in low light conditions. This was observed under the Fanning Springs Bridge, as noted in Table 1, where the lighting conditions were too dark for the aircraft to utilize the optical flow camera for reliable navigation. After the flight, the camera was optimized for low light conditions, but improvements could still be made such as adding external light sources to the aircraft or utilizing other types of EO sensors.

Some other noteworthy improvements that were made as a direct result of the bridge inspections included decreasing the focal length of the camera. It was found that the aircraft had trouble tracking the bottom of the Atlantic Boulevard Bridge, especially at closer distances. This was due to the flow rates exceeding the limits of the camera in conjunction with lack of feature points. By switching to a lower focal length, the field of view was increased, thus increasing the number of focal points and reducing the effective flow rates. This switch resulted in better position hold performance of the aircraft.

Wind gusts would also degrade the altitude tracking performance of the aircraft, causing the aircraft to change altitude. These altitude changes were due to a venturi effect where the airflow over the body of the aircraft would cause a pressure change. This pressure change would cause the barometer to read out an erroneous altitude. The first attempt to solve this was to include foam around the sensor to baffle the wind, resulting in little success. The problem was finally overcome with the development of the hybrid altitude algorithm, which fuses laser rangefinder measurements with the barometer readings which were not affected by wind.

The final deficiency that was found was when the aircraft is flown near the edges of a bridge, the position hold performance would degrade. This was found to be caused by the optical flow camera field of view capturing parts of the sky. After this discovery, the aircraft was positioned such that it would keep a prescribed distance from the edges of a bridge. Further testing would be needed to integrate a system to automatically switch between GPS navigation and optical flow when flying near the edges of a bridge.

## 7. Discussion

Comparing the performance between the DJI system to that of the UF aircraft, it was found that the hybrid altitude controller incurred a reduction in altitude variance by a factor of 7. In addition, the optical flow based navigation system produced a tighter lateral position hold than the GPS-based DJI system. This is in part due to the increased update rate of the optical flow-based navigation system which can output at 400 Hz as compared to typical GPS offerings which update between 5 and 10 Hz for COTS receivers. Also, optical flow systems are not affected by interference such as multipathing, thus reducing the possibility of glitching, but at the tradeoff of requiring a sufficiently textured surface. This improved position hold performance allows the operator to navigate around a cluttered environment with better precision than a COTS system in addition to having an overhead obstacle avoidance algorithm that helps keep the vehicle at a prescribed standoff distance.

After this study was completed, DJI released a commercial drone system with a similar use case as the one proposed here. While it did address a majority of the challenges proposed by under bridge deck inspections such as employing overhead optical flow and an above mounted camera system, there were a few areas where our proposed system excelled. The camera system utilized for the proposed UAS has superior image and video quality as compared to the Matrice 300 RTK (DJI, Nanshan, Shenzhen, China) as we are utilizing a 24 MP full size mirrorless camera with a shutter. This allows for better detail and superior exposure control which is necessary for low light conditions as typically seen under the bridge. In addition, our proposed aircraft has a greater vertical avoidance range of 120 m as compared to the 40 m from the Matrice 300 RTK. Position hold performance comparison could not be done at this time, but due to the HDR capability of the camera used for the proposed UAS system, the optical flow system should be more robust than the Matrice 300 RTK. In addition, due to the availability of COTS components and the simplicity of the system, our system should also cost less than the Matrice 300 RTK.

One thing to note though, is that while the system is able to hold a stationary position, the optical flow-based system needs to be calibrated for inaccuracies in the focal length. For example, while both the linear fit of the optical flow system velocity versus actual velocity as shown in Figure 18, unlike the DJI system, the line does not have a unitary slope. To correct for this, the focal rate should be scaled such that this fit line approaches one.

For conducting bridge inspections, the following procedure was utilized to optimize the speed and efficiency. Since the aircraft has vertical obstacle avoidance utilizing a single, narrow beam laser, the vehicle should travel parallel to the bridge girders. This would result in the aircraft following the curve of the bridge deck while minimizing changes in altitude from traversing perpendicular to the girders themselves. Another benefit of flying between the girders is the aircraft can fly closer to the bridge deck for a given standoff distance, thus allowing for closer views of the bridge defects. For the inspections, the standoff distance was adjusted so that the distance exceeds the depth of the bridge girder. This was done so to prevent the aircraft from impacting the side of the girder. To fly at closer distances, the aircraft would need to be equipped with either propeller guards or automatic horizontal obstacle avoidance [8]. Due to the limitation of utilizing the bridge deck itself for navigation utilizing the optical flow camera, flying along the outer most girder was avoided since parts of the sky would affect the optical flow readings in a negative way. This was not an issue since the aircraft could fly with GPS along the sides of the bridge. Lastly, the aircraft would approach the bridge deck from the center of the bridge section before the optical flow algorithm locks in. This is to prevent the aircraft from contacting the bridge piers before the aircraft can hold position on its own. The inspection would proceed with the aircraft flying up and down the area between bridge girders until the whole area is covered with pauses to collect photographs of possible defects.

## 8. Conclusions

The results presented in this paper demonstrate that the proposed UAS can maintain position hold in a GPS-denied environment with performance superior to a standard COTS system using GPS. In addition, the hybrid altitude algorithm was also shown to improve altitude hold performance in a GPS-denied environment as compared to a barometric pressure base system. While the system was able to be utilized for a number of real-world bridge inspections, there are still areas for further improvement. The first is improving the reliability of switching between GPS navigation and the GPS-denied algorithm as the system requires a full view of the bottom of the bridge deck and would struggle at the edges of the bridge where only a part of the bridge deck was in full view. In addition, stereovision can be explored for further performance improvements as it can generate individual depths for each feature point instead of the current proposed algorithm which averages all the feature points and assumes a uniform standoff distance. The utilization of this proposed UAS presents a novel approach which may ultimately enable unskilled pilots to successfully navigate the inspection of bridges and other structures enhancing safety and reducing cost.

## Figures and Tables

**Figure 1 sensors-20-05358-f001:**
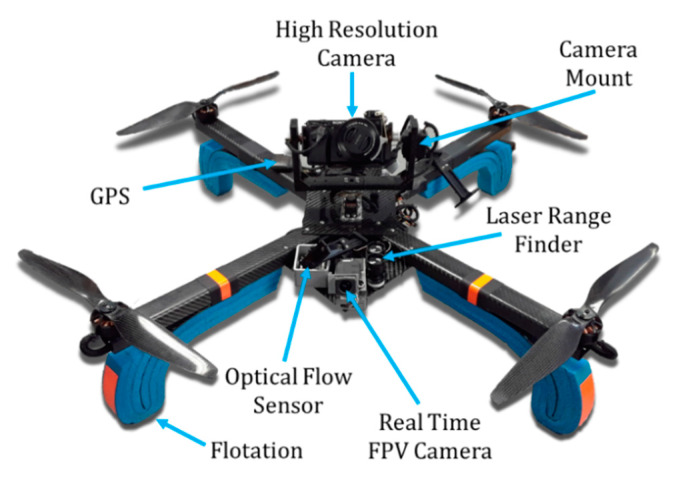
Proposed hardware configuration of bridge inspection UAS.

**Figure 2 sensors-20-05358-f002:**
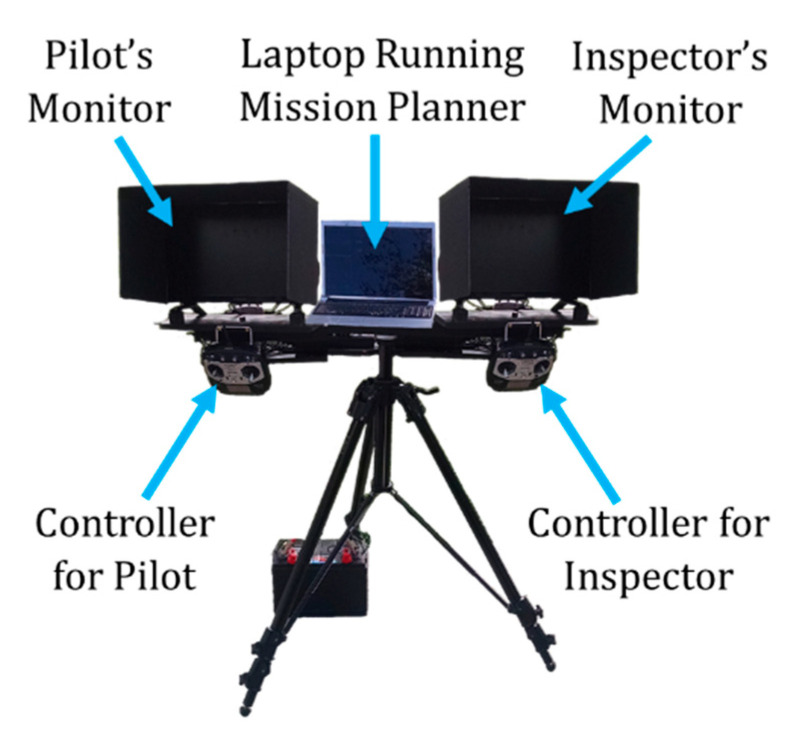
Ground Station used for proposed UAS.

**Figure 3 sensors-20-05358-f003:**
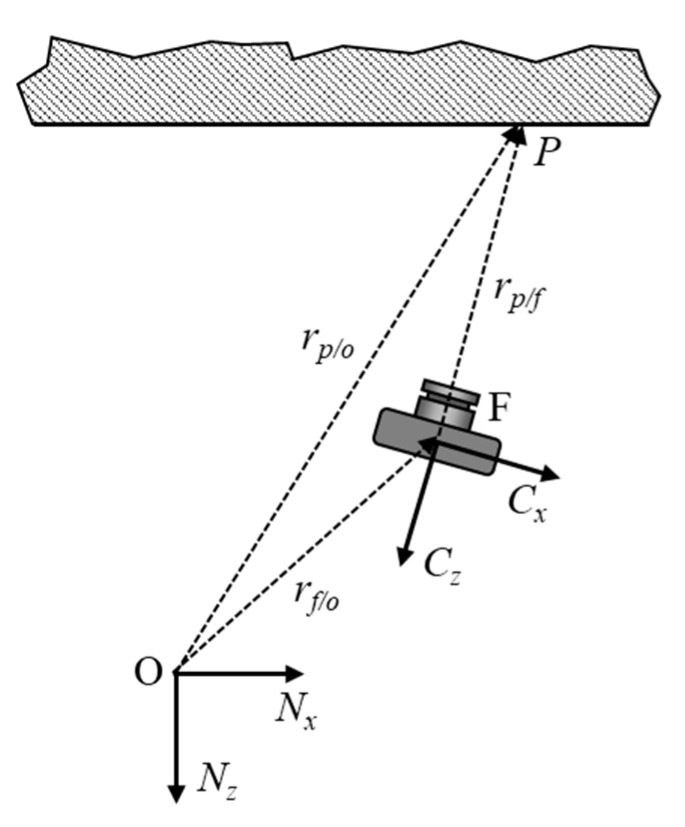
Relationship between the camera focal point, the origin, and the optical axis projection onto the surface above.

**Figure 4 sensors-20-05358-f004:**
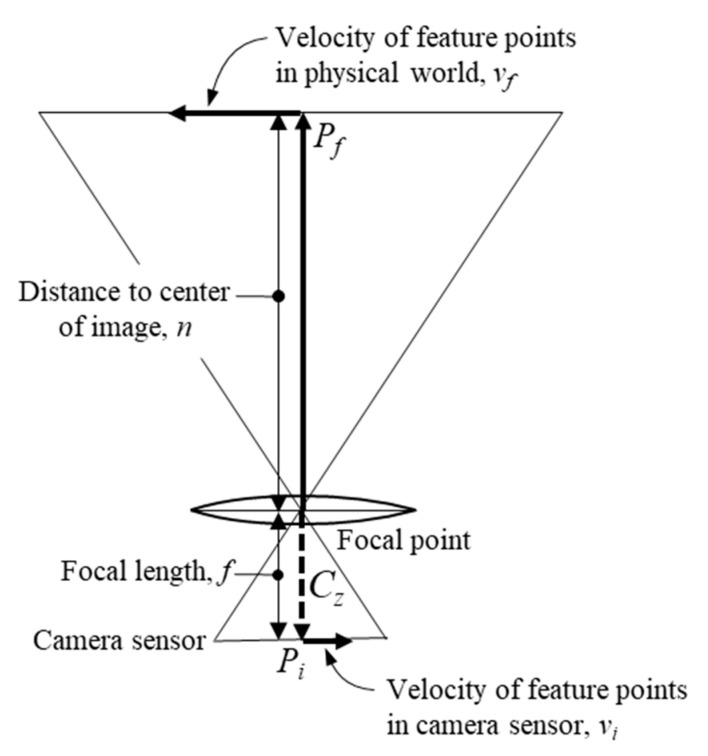
Pinhole model of a camera used for optical flow derivation.

**Figure 5 sensors-20-05358-f005:**
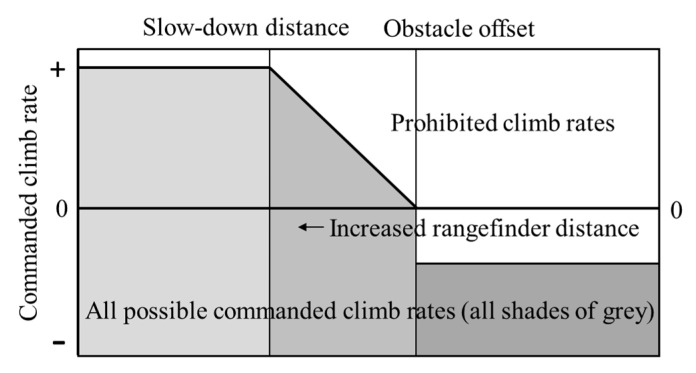
Allowable climb rates by pilot based off overhead standoff distance.

**Figure 6 sensors-20-05358-f006:**
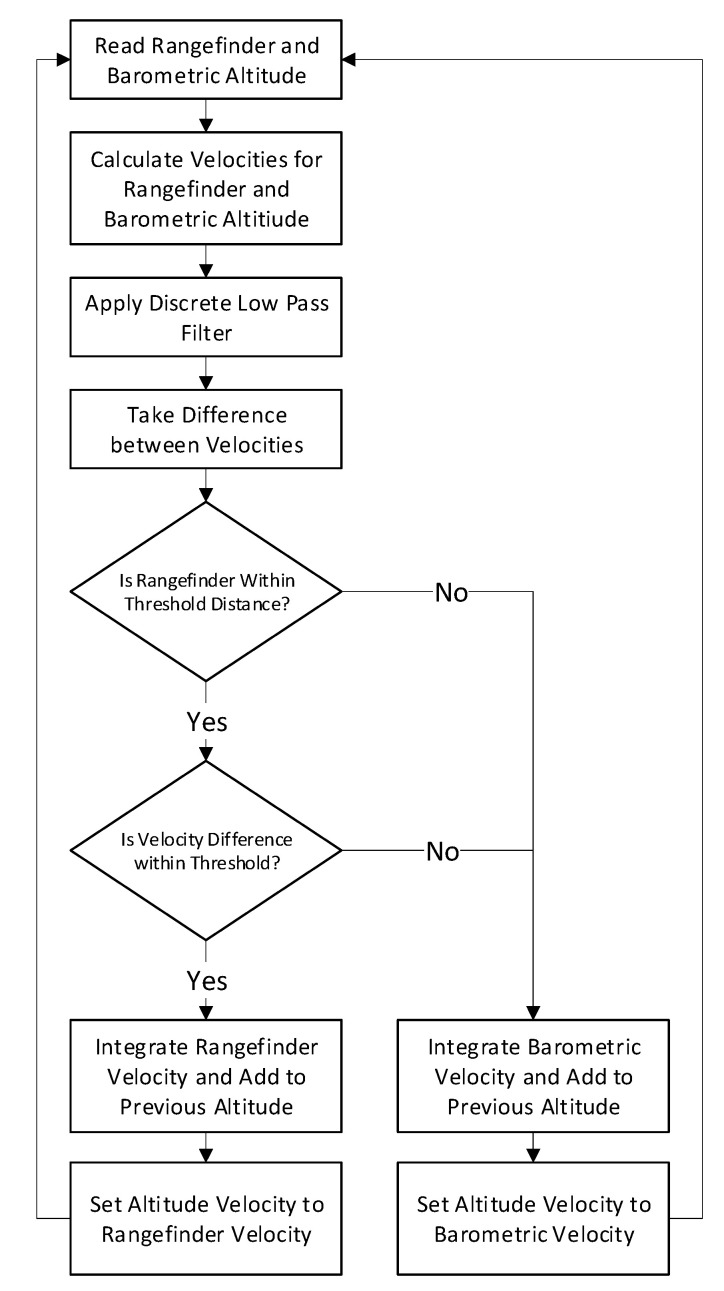
Hybrid altitude state machine.

**Figure 7 sensors-20-05358-f007:**
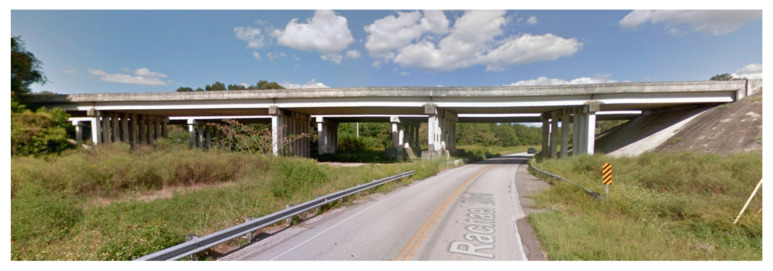
Google Maps Street View of the bridge used for development of UAS system.

**Figure 8 sensors-20-05358-f008:**
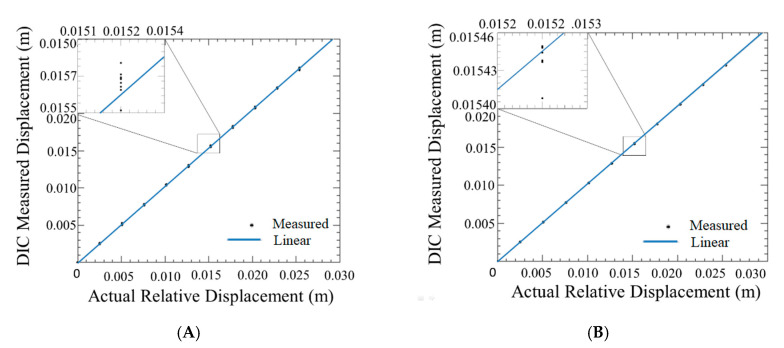
DIC measurement resolution results: (**A**) Out-of-plane direction; (**B**) In-plane direction.

**Figure 9 sensors-20-05358-f009:**
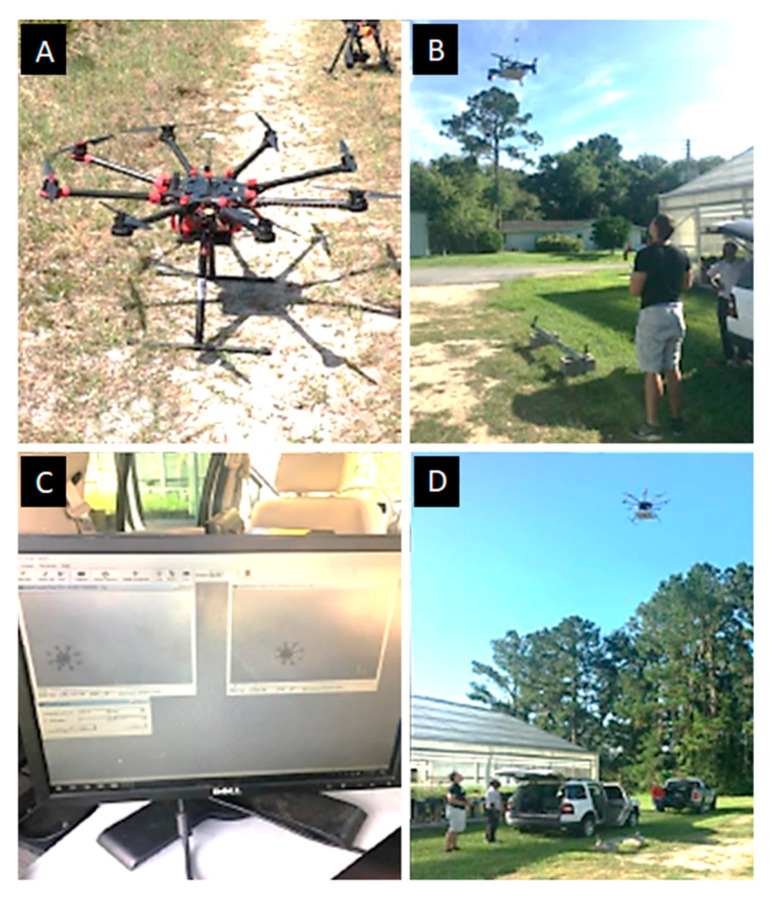
(**A**) The DJI S100+ COTS drone, (**B**) position hold test of the bridge inspection drone, (**C**) DIC image capture computer showing both camera images, (**D**) position hold test of the S1000+ drone.

**Figure 10 sensors-20-05358-f010:**
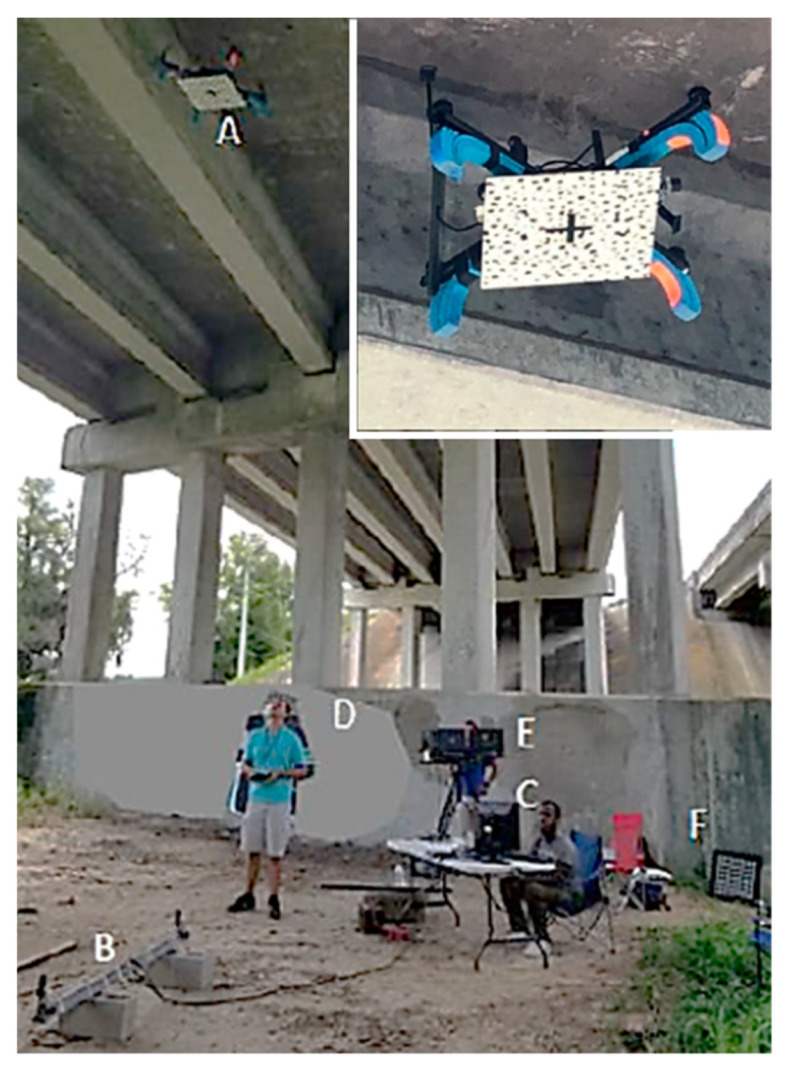
Experimental setup under the US 441 bridge, Alachua, FL. The photograph shows the drone (**A**), DIC stereo cameras mounted on a beam (**B**), computer for DIC image acquisition(**C**), pilot (**D**), ground station (**E**), and DIC calibration grid (**F**).

**Figure 11 sensors-20-05358-f011:**
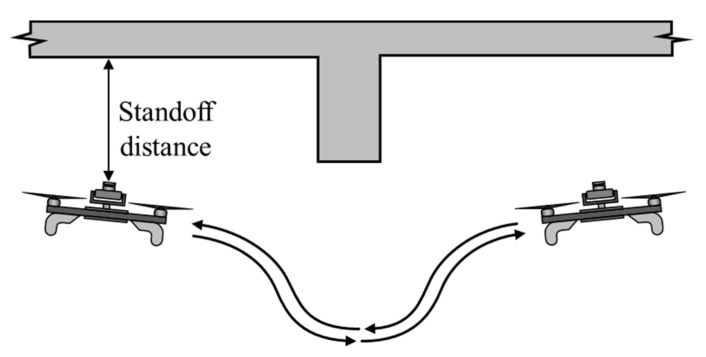
Trajectory of aircraft during the vertical obstacle avoidance test.

**Figure 12 sensors-20-05358-f012:**
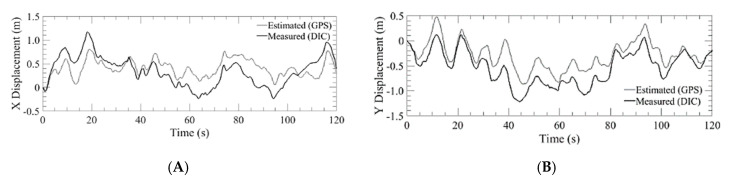
Lateral displacement estimated by GPS of the DJI S1000+ versus the measured value of the DIC for run 2 where: (**A**) X displacement; (**B**) Y displacement.

**Figure 13 sensors-20-05358-f013:**
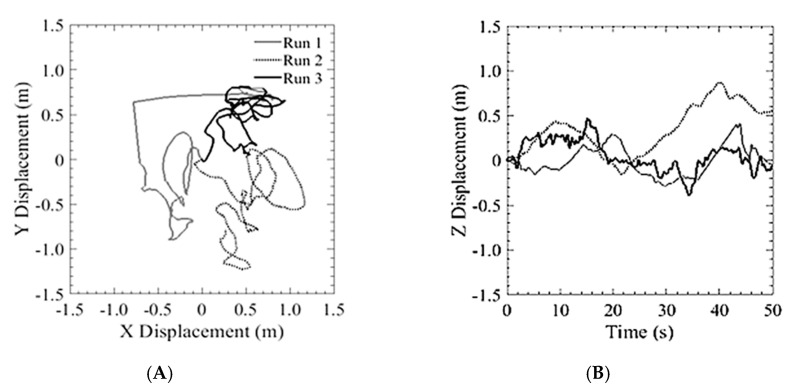
Typical performance of a COTS GPS based system: (**A**) Lateral position hold; (**B**) Vertical position hold.

**Figure 14 sensors-20-05358-f014:**
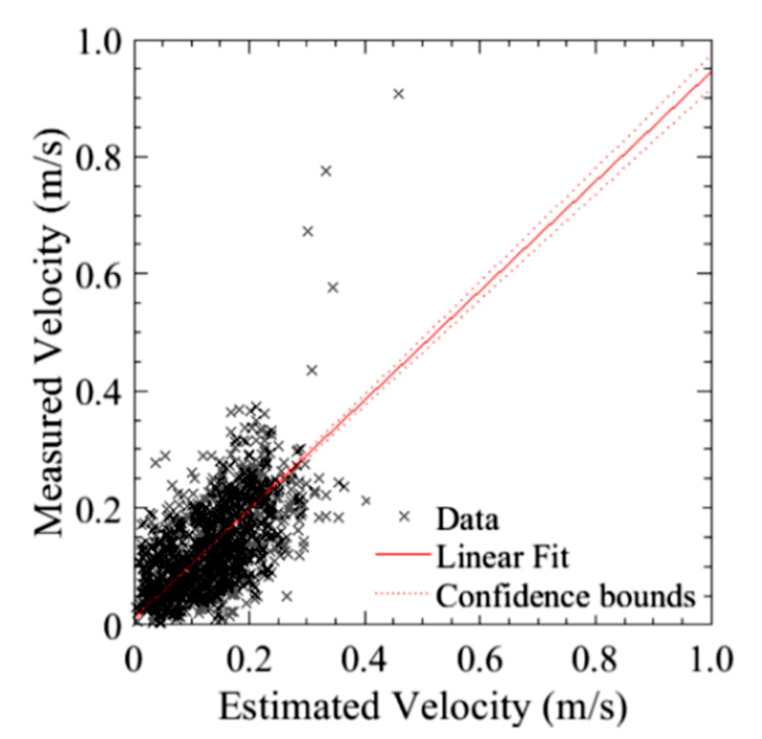
Estimated velocity (from GPS) versus measured velocity (DIC) of the best run for the S1000+.

**Figure 15 sensors-20-05358-f015:**
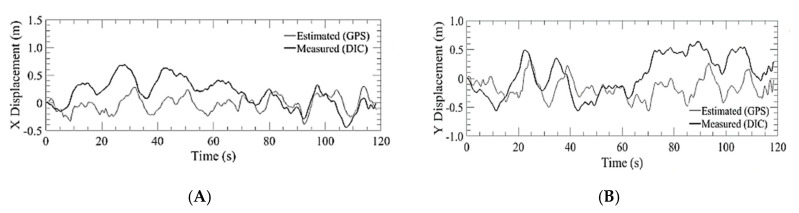
Lateral displacement estimated by GPS of the bridge inspection airframe versus the measured value of the DIC for run 3 where: (**A**) X displacement; (**B**) Y displacement.

**Figure 16 sensors-20-05358-f016:**
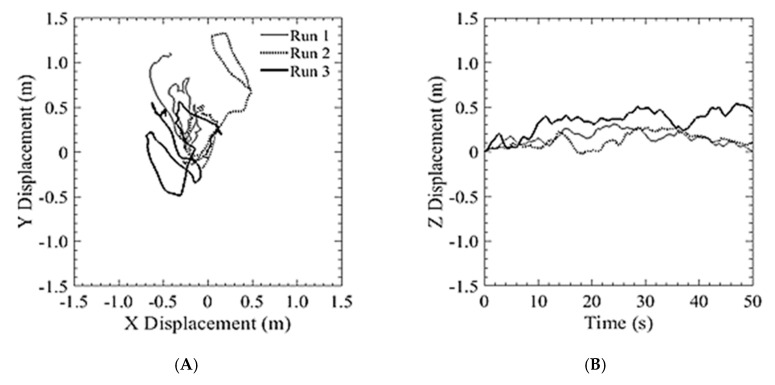
Position hold performance of GPS on bridge inspection airframe: (**A**) Lateral position hold; (**B**) Vertical position hold.

**Figure 17 sensors-20-05358-f017:**
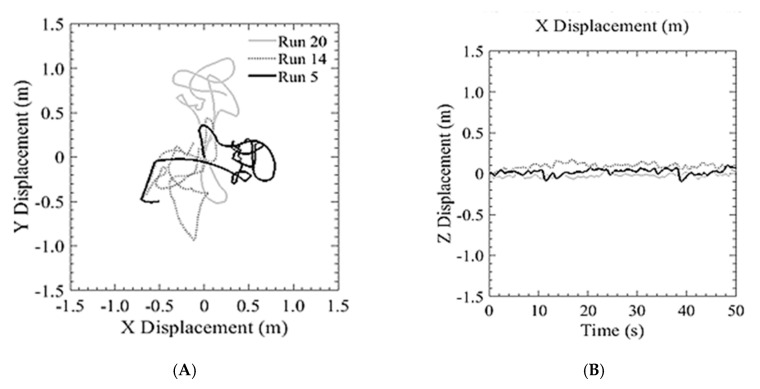
Position hold performance of optical flow and hybrid altitude controller: (**A**) Lateral position hold; (**B**) Vertical position hold.

**Figure 18 sensors-20-05358-f018:**
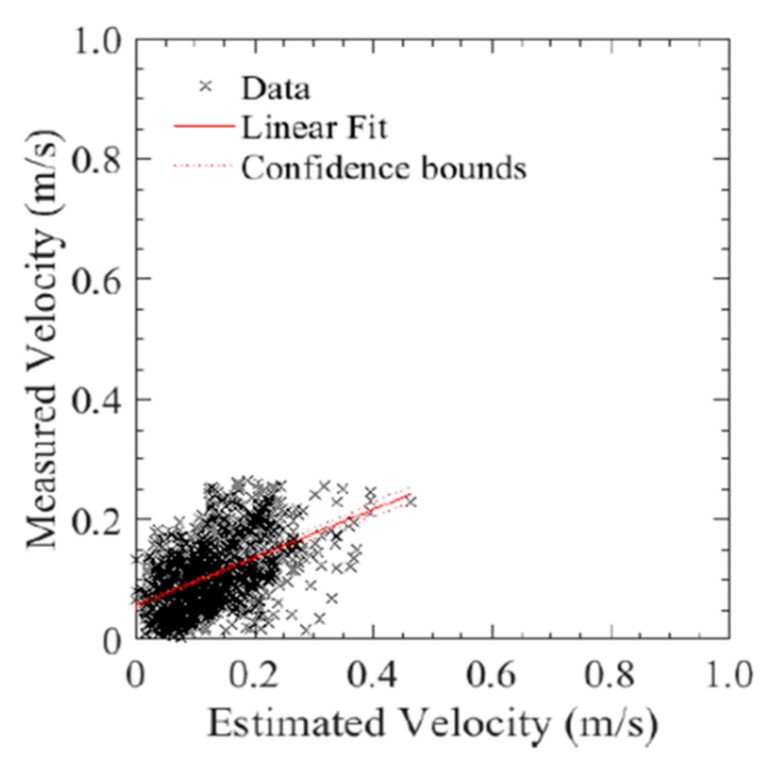
Estimated velocity (from GPS) versus measured velocity (DIC) of the best run for the bridge inspection airframe.

**Figure 19 sensors-20-05358-f019:**
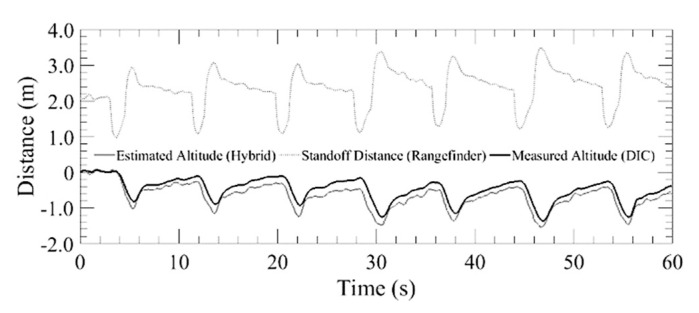
Standoff distance v. aircraft altitude for overhead obstacle avoidance test.

**Figure 20 sensors-20-05358-f020:**
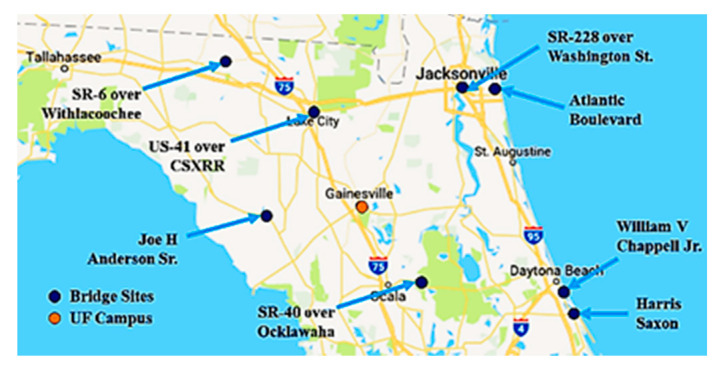
Locations of bridges inspected by UF aircraft (reprinted with permission from Tomiczek, A. *Locations of bride inspections*. Masters dissertation. University of Florida: Gainesville, FL, USA, 2018 p. 61, Figure 1, Figure 2, Figure 3 and Figure 4).

**Table 1 sensors-20-05358-t001:** Summary of bridges inspected alongside Florida Department of Transportation and modifications as a result of outcome.

Bridge	Type	Challenges	Resulting Modifications
Atlantic Blvd Bridge, Jacksonville, FL	Steel and Concrete	-Poor position holding from optical flow. -Collision from bridge underdeck	-Optical flow camera focal length reduced -Automatic flight mode switching implemented -Flotation material added to airframe
Donlawton Bridge, Port Orange, FL	Steel and Concrete	-Shimmer from water observed on edge of bridge	-No negative impacts in performance
South Causeway Bridge, New Smyrna, FL	Prestressed Concrete	-Trouble holding position on edge of bridge	-Avoid optical hold flights on edge of bridge
Suwannee River Bridge, Fanning Springs, Fl	Prestressed Concrete	-Optical flow algorithm could not hold position	-Optimized optical flow camera for low light areas -increased optical flow camera FOW
Withlacoochee River Bridge, Route 6, FL	Steel	-None	-None
Ocklawaha River Bridge, Route 40, FL	Steel and Prestressed Concrete	-Trouble holding position for steel beams	-Flying at lower standoff distance
NW Main Blvd Bridge, Lake City, FL	Steel	-Altitude variance from wind	-Development of rangefinder based altitude estimation
Hart Bridge Expressway, Jacksonville, FL	Steel and Prestressed Concrete	-Altitude variance from wind -Trouble holding position for steel beams	-Development of rangefinder based altitude estimation

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
