# Peer review of "Design of a Small Unmanned Aircraft System for Bridge Inspections"

_sensors, 2020, doi:10.3390/s20185358_

Round 1

Reviewer 1 Report

The paper describes the efforts on an experiment that appears to be comprehensive and commendable. But there are some issues that could be addressed to improve the study and paper. Some specific comments are as follows:

  1. The ground station (two controllers (pilot and inspector) and two monitors plus a laptop on a tripod) appears to be cumbersome for practical implementation; it may be okay for the research purpose.
  2. The overall approach for the Vertical Obstacle Avoidance (Section 3.2) and Hybrid Altitude Control (Section 3.3) are discussed but the technical algorithms are not well described. There are only a few references on the technical algorithms.
  3. Is there any comparable commercial drone package (that can perform the same duties) equivalent to the assembly developed here?
  4. How comparable are the efforts by other transportation agencies which are utilizing or planning to use drones for bridge inspection?
  5. The contents of Table 1 appear to be repeated by error at different positions on the paper.

Author Response

The paper describes the efforts on an experiment that appears to be comprehensive and commendable. But there are some issues that could be addressed to improve the study and paper. Some specific comments are as follows:

1. The ground station (two controllers (pilot and inspector) and two monitors plus a laptop on a tripod) appears to be cumbersome for practical implementation; it may be okay for the research purpose.

We agree that the ground station in its current form is bulky, but for a commercial application it will need to be produced in a much smaller form factor. A note was made in Lines 128-129 in the revised document.

2. The overall approach for the Vertical Obstacle Avoidance (Section 3.2) and Hybrid Altitude Control (Section 3.3) are discussed but the technical algorithms are not well described. There are only a few references on the technical algorithms.

I do agree that the references are not sufficient for the vertical obstacle avoidance algorithm in addition to the Hybrid Altitude controller. We have thus included the equation derivation of the Kalman filter used for the standoff distance algorithm in Lines 192-203 in addition to going into more detail for the hybrid altitude algorithm (see lines 244-250).

3. Is there any comparable commercial drone package (that can perform the same duties) equivalent to the assembly developed here?

At the time of testing of the aircraft 2 years ago, there was not a comparable commercial drone package being offered. Since then the DJI Matrice 300RTK has been released and has proven to have similar characteristics such as employing overhead optical flow and an option for mounting a camera above the air vehicle. Even with these improvements and capabilities our system still has a superior camera system with a 24MP Mirrorless DSLR while having exceptional exposure range and automatic control which are all needed for capturing defects under the bridge. In addition our GPS denied navigation system should have the superior performance for lower light conditions from utilizing an optical flow camera employing HDR and a laser rangefinder for depth reconstruction with 120m range (as compared to 40m from the DJI Matrice). This comparison has been included in Lines 511-523.

4. How comparable are the efforts by other transportation agencies which are utilizing or planning to use drones for bridge inspection?

At the time of testing of the UAS, a majority of the efforts of other transportation agencies consisted of flying COTS drones without the use of GPS under the bridge by relying on an skillset of an experienced drone pilot to fly the aircraft manually while others would only fly to the side of the bridge and try to view defects at a distance. There were many recommendations to implement various forms of GPS denied navigation into the UAS, but none actually tested these implementations on an actual bridge inspection. References to these studies were included in lines 43-45.

5. The contents of Table 1 appear to be repeated by error at different positions on the paper.

We agree that was an error and have thus been removed.

Please see the attached word document of the tracked changes to the manuscript.

Reviewer 2 Report

This paper proposes a method using UAVs for bridge inspection, the method are potentially interesting and worthy of eventual publication.

Although the content is sufficient, the following points may make the paper more readable to readers.

  • Which term, UAS or UAV, is more common and appropriate for this paper?
  • How much of the environment is GPS unavailable? This will provide evidence of the utility of this study, so please show any data you have.

Author Response

This paper proposes a method using UAVs for bridge inspection, the method are potentially interesting and worthy of eventual publication.

Although the content is sufficient, the following points may make the paper more readable to readers.

1. Which term, UAS or UAV, is more common and appropriate for this paper?

For this paper, UAS would be more appropriate as we were developing not just the air vehicle itself, but also the ground station as well to create an optimal complete system for bridge inspection.

2. How much of the environment is GPS unavailable? This will provide evidence of the utility of this study, so please show any data you have.

The areas that are GPS denied are any of the areas directly below the bridge deck. While at some bridges, the geometry does allow for a sufficient GPS signal, there is the possibility of receiving GPS multipathing, thus causing an incorrect position output from the GPS receiver itself. This multipathing can result in uncommanded movements of the air vehicle which can lead to collision with the cluttered environment, thus why anywhere under the bridge deck should be treated as a GPS denied environment. I have added references to studies that have observed these GPS denied zones under the bridge in line 43.

See attached word document for the manuscript with the tracked changes.

Reviewer 3 Report

  1. The authors cited only four papers including one of their own published recently to provide background to this problem. Many state department of transportation have been experimenting UAV for bridge inspection. Have they encountered the problem this paper attempts to address?
  2. There is commercial UAV currently on the market for example DJI Matrice 300RTK that is specifically designed for inspection purpose. The authors should compare their system to DJI Matrice in terms of performance improvement and cost advantage.  
  3. Those equations look a little vague, correct if possible.
  4. Text in line 425, detailed in... (Table...)?
  5. All tables do not have captions except for the last one, please correct. 

Author Response

1. The authors cited only four papers including one of their own published recently to provide background to this problem. Many state department of transportation have been experimenting UAV for bridge inspection. Have they encountered the problem this paper attempts to address?

We do agree there are more papers addressing this problem from other transportation agencies, and have made references to them in Lines 44- 46.

2. There is commercial UAV currently on the market for example DJI Matrice 300RTK that is specifically designed for inspection purpose. The authors should compare their system to DJI Matrice in terms of performance improvement and cost advantage.

At the time of testing (2 years ago) there were not systems comparable to the one proposed here. Now that DJI has just released the Matrice 300RTK, we have included a comparison in lines 511-523.

3. Those equations look a little vague, correct if possible.

We do agree that some of the equations are a little vague, thus we have included a more detailed analysis of the proposed equations to help walk the reader through the derivation at lines 151-152, lines 156-158, and Line 164.

4. Text in line 425, detailed in... (Table...)?

We agree that was an error and have thus removed the duplicate table.

5. All tables do not have captions except for the last one, please correct. 

We agree that was an error and have thus removed the duplicate tables that do not have a caption.

See the attached word document of the manuscript for tracked changes.
